# Prevalence of undernutrition and associated factors among adults taking antiretroviral therapy in sub-Saharan Africa: A systematic review and meta-analysis

**Awole Seid**[1,2]*, **Omer Seid**[3], **Yinager Workineh**[4], **Getenet Dessie**[1,5], **Zebenay Workneh Bitew**[2,6]

**1** Department of Adult Health Nursing, College of Medicine and Health Sciences, Bahir Dar University, Bahir Dar, Ethiopia, **2** Center for Food Science and Nutrition, Addis Ababa University, Addis Ababa, Ethiopia, **3** Department of Nutrition and Dietetics, School of Public Health, College of Medicine and Health Sciences, Bahir Dar University, Bahir Dar, Ethiopia, **4** Department of Pediatrics and Child Health Nursing, College of Medicine and Health Sciences, Bahir Dar University, Bahir Dar, Ethiopia, **5** National Centre for Epidemiology and Population Health, College of Health and Medicine, Australian National University, Canberra, Australian Capital Territory, Australia, **6** St. Paul Millennium Medical College, Addis Ababa, Ethiopia

* sawlayehu@gmail.com

**Data Availability Statement:** All relevant data are within the paper and its Supporting Information files.

## Abstract

### Background

Undernutrition (Body Mass Index < 18.5 kg/m2) is a common problem and a major cause of hospital admission for patients living with HIV. Though sub-Saharan Africa is the most commonly affected region with HIV and malnutrition, a meta-analysis study that estimates the prevalence and correlates of undernutrition among adults living with HIV has not yet been conducted. The objective of this study was to determine the pooled prevalence of undernutrition and associated factors among adults living with HIV/AIDS in sub-Saharan Africa.

### Methods

Studies published in English were searched systematically from databases such as PubMed, Google Scholar, and gray literature, as well as manually from references in published articles. Observational studies published from 2009 to November 2021 were included. The data extraction checklist was prepared using Microsoft Excel and includes author names, study area, publication year, sample size, prevalence/odds ratio, and confidence intervals. The results were presented and summarized in accordance with the Preferred Reporting Items for Systematic Reviews and Meta-analyses (PRISMA) standard. Heterogeneity was investigated using the Q test, $I^2$, $\tau^2$, $\tau$ and predictive interval. STATA version 17 was used to analyze the data. A meta-analysis using a random-effects model was used to determine the overall prevalence and adjusted odds ratio. The study has been registered in PROSPERO with a protocol number of CRD42021268603.

**Funding:** The authors received no specific funding for this work.

**Competing interests:** The authors have declared that no competing interests exist.

## Results

In this study, a total of 44 studies and 22,316 participants were included. The pooled prevalence of undernutrition among adult people living with HIV (PLWHIV) was 23.72% (95% CI: 20.69–26.85). The factors associated with undernutrition were participants' age (AOR = 0.5, 95% CI: 0.29–0.88), gender (AOR = 2.08, 95% CI: 0.22–20.00), World Health Organization (WHO) clinical stage (AOR = 3.25, 95% CI: 2.57–3.93), Cluster of Differentiation 4 (CD4 count) (AOR = 1.94, 95% CI: 1.53–2.28), and duration of ART (AOR = 2.32, 95% CI: 1.6–3.02).

## Conclusion

The pooled prevalence of undernutrition among adult PLWHIV in sub-Saharan Africa remained high. WHO clinical stage, CD4 count, duration of ART treatment, age, and sex were found to be the factors associated with undernutrition. Reinforcing nutrition counseling, care, and support for adults living with HIV is recommended. Priority nutritional screening and interventions should be provided for patients with advanced WHO clinical stages, low CD4 counts, the male gender, younger age groups, and ART beginners.

## Introduction

According to the United Nations Program on HIV/AIDS (UNAIDS) 2021 fact sheet, there were 38.4 million people living with HIV, of whom 36.7 million were adults (15 years of age or older). Seventy-six percent of adults living with HIV had access to ART treatment [1]. Sub-Saharan Africa accounted for 57% of all new HIV infections in 2019 [2]. Similarly, Eastern and Southern Africa constituted the largest number of AIDS-related deaths (280,000) globally in 2021 [1]. Despite progress in HIV care, the overall life expectancy among adults living with HIV is 5 to 10 years less than that of uninfected adults [3]. Malnutrition takes the lion's share in increasing the risk of mortality and the occurrence of other opportunistic infections among adults living with HIV [4, 5].

Poor nutrition and HIV have bidirectional relationships and exacerbate one another [6]. In resource-limited countries like sub-Saharan Africa, many people living with HIV (PLWHIV) on long-term ART follow-up lack adequate nutrition [7, 8], and, undernutrition is an indicator of a poor prognosis of HIV care [9]. Similarly, malnutrition is among the major causes of hospital admission in PLWHIV [10]. Meanwhile, HIV affects nutritional status in three distinct ways. It decreases food consumption (through poor appetite and inability to eat and swallow), raises energy needs (up to 20% more energy), and hinders the body's ability to absorb nutrients. All these factors predispose patients to undernutrition and finally to wasting syndrome [11]. Thus, identifying and treating malnutrition in people living with HIV can fasten recovery from infection, enhance immunity, and possibly slow the progression to AIDS [12].

A meta-analysis study in Ethiopia showed the pooled prevalence of undernutrition among adults receiving ART was 26%. Undernutrition among people living with HIV is associated with socio-demographic and clinical factors such as age, WHO clinical stage, CD4 count, duration of ART treatment, and food security [13, 14]. There is a need to estimate the overall prevalence of undernutrition in Sub-Saharan Africa, as the region remains the world's epicenter of HIV transmission. Similarly, several studies indicated that the region is affected by food insecurity, which is directly linked to undernutrition for individuals, households, and communities affected by HIV [15–18]. Beside this, several nutritional programs to address undernutrition in adults living with HIV in sub-Saharan Africa have been instituted without

consolidated evidence on the overall estimates of the prevalence and correlates of undernutrition [19]. Moreover, the available primary studies in sub-Saharan Africa lack consistency and are not conclusive. Therefore, the purpose of this meta-analysis study was to determine the pooled (overall) prevalence of undernutrition and its associated factors among adults living with HIV in sub-Saharan Africa. This study will help policymakers devise evidence-based nutrition intervention programs for patients living with HIV in sub-Saharan Africa.

## Methods

### Inclusion and exclusion criteria

Both published and unpublished observational studies (i.e., cross-sectional, case-control, and cohort) conducted among HIV-positive adults in SSA countries were included. Articles published only in the English language were included. On the other hand, studies with no free full texts, and studied only pregnant women were excluded. For clarity, the topic is described using a PICO format as follows:

- **Population (P):** Adult patients ($> 15$ year, as defined by UNAIDS and other studies) [20].

- **Intervention or exposure (I):** Living with HIV

- **Comparison or control (C):** Living without HIV

- **Outcome (O):** Undernutrition (prevalence)

### Information source and search strategy

We used the databases mainly PubMed, Google Scholar, and Gray (unpublished) literature, university repositories, and manual searches of references from a list of included articles. Articles published from 2009 to November 10, 2021, were included. We used 2009 as there was a previous study among women living with HIV and published in 2008. However, our study also included male adults. Articles identified through the electronic searches were exported and managed using EndNote Version 8 reference manager. Articles from PubMed were accessed using the following keywords (**Table 1**).

### Data collection process and data items

The authors prepared data extraction using Microsoft Excel. All relevant data for this review were extracted by two reviewers (AS and OS). The disparities between reviewers at the time of data abstraction were resolved through discussion with the third author (ZWB).

**Table 1. The search strategy used in the PubMed database, 2021.**

| Key variables | Searching words in PubMed |
|---|---|
| **Prevalence of undernutrition** | "Proportion" or "Prevalence" or "Magnitude" or "Burden" AND "Malnutrition" OR "Undernutrition" OR "Under-weight" OR "Wasting" OR "Malnourished" AND "Adult" AND "Living with HIV" OR "HIV-positive" OR "HIV-infected" OR "Anti-Retroviral Therapy" AND (Each country) sub-Saharan Africa". |
| **Associated factors** | "Associated factors" OR "Determinants" OR "Predictors" OR "Correlates" AND "Malnutrition" OR "Undernutrition" OR "Under-weight" OR "Wasting" OR "Malnourished" AND "Adult" AND "Living with HIV" OR "HIV-positive" OR "HIV-infected" OR "Anti-Retroviral Therapy AND (Each country) in sub-Saharan Africa". |

Studies were selected after two reviewers (AS) and (OS) independently screened for inclusion eligibility. A third author (GD) was involved in resolving disagreements between the two authors.

The data extraction sheet included primary authors, publication year, country, study design, sample size, response rate, study setting, study population, proportion, 95% confidence interval, and the logarithm of proportion S1 Table.

## Effect measures

We include studies that measure under-weight (undernutrition) using BMI < 18.5 Kg/m$^2$. Underweight was utilized as an indicator of advanced malnutrition, despite the fact that it does not reliably indicate the nutritional status of adults [21]. The proportion of undernutrition was calculated by dividing the number of individuals under-nourished by the total sample of study subjects included in the final analysis. We used the adjusted odds ratio (AOR) as an effect measure to find associated factors of undernutrition.

## Risk of bias assessment

Newcastle Ottawa Scale (NOS) adapted for cross-sectional studies was used to assess the quality of the studies. NOS has three categories and has a maximum score of 10 for cross-sectional studies. The categories are selection (maximum of 5 stars), comparability (maximum of 2 stars), and study outcome (maximum of 3 stars). Each study was independently appraised by two authors. Disagreements between authors were resolved through discussion with a third author. Finally, the quality score of each study was calculated as the sum of scores, thus ranging from zero to ten for cross-sectional studies, and zero to nine for cohort and case-control studies. A score of greater or equal to 6 points was considered "good" and included in the study [22]. Additionally, publication bias was assessed using Egger's regression test, funnel plot, and sensitivity analysis.

## Synthesis methods

Data analysis was performed using STATA (version 17) software. We employed a random effect model to find the pooled prevalence and associated factor estimates of under-nutrition. Heterogeneity of effect sizes was assessed using I$^2$, $\tau^2$,$\tau$ and prediction interval. Subgroup and trim and fill were performed to deal with the potential source of heterogeneity. Sensitivity analysis was also performed. The Preferred Reporting Items for Systematic Reviews and Meta-Analyses (PRISMA) checklist is used for data presentation S2 Table [23].

# Results

## Study selection and characteristics

A total of 3,477 articles were identified from PubMed, Google scholar, and gray literature in the initial search. After the removal of 1,220 duplicates, 2,257 articles were screened for title and abstract. In the next step, 2,194 articles were excluded based on titles and abstracts. The full texts of 62 articles were downloaded and assessed against inclusion criteria. Thus, 18 articles were excluded for the following reasons: 7 studies did not report data on the outcome variable [5, 24–29], six studies were review papers [4, 14, 30–33], three studies focused on children [30, 34, 35], one study was conducted out of SSA [36], one study focused on pregnant women [37], and one study lacks full text [38]. Finally, 44 studies were included in final systematic review and meta-analysis (**Fig 1**) [23]. No studies were excluded after appraising the quality using NOS.

In this study, a total of 22,316 adults living with HIV were included. The sample size of the included studies ranged from 145 in Botswana to 3,993 in Tanzania. Of the 44 included studies, only one was a retrospective cohort and the rest were carried out using cross-sectional

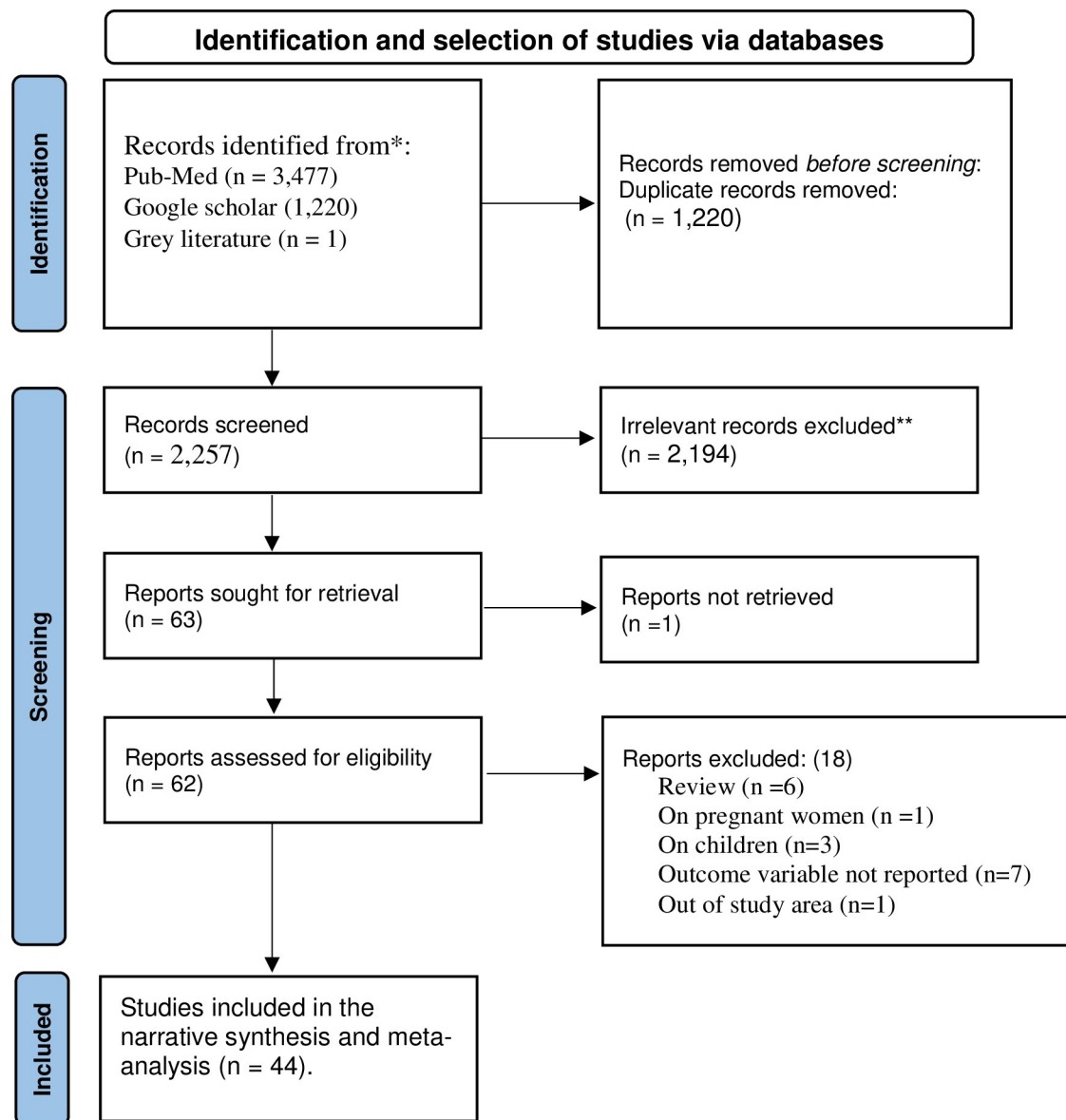

**Fig 1. Flow chart showing the sequence of study selection on undernutrition among adult PLWHIV in sub-Saharan Africa, 2009–2021.**

study designs. The largest number of articles (33 studies) were reported from Ethiopia [8, 39–70] and only one article is reported from Kenya [71], Botswana [72], Ghana [73], Democratic Republic of Congo (DRC) [74], Senegal [75], Uganda [76], and Zimbabwe [77]. Four articles were from each of South Africa [78, 79] and Tanzania [13, 80]. Regarding the publication year, studies were published from 2009 to 2021 and the majority of the studies (11), were reported in 2020, followed by nine in 2017, and five in each of 2018 and 2015 **(Tables 2 and 3).**

## Quality appraisal results

The heterogeneity test of the study revealed $I^2$ = 96.8%, $\tau^2$ = 98.83, $\tau$ = 9.94, prediction interval (13.8–33.68%), and 95% confidence interval of the average estimate (20.69–26.79%). The

**Table 2. Study characteristics included in meta-analysis of undernutrition among adult PLWHIV in sub-Saharan Africa, 2009–2021.**

| S.N | Authors | Publication Year | Country | Sample Size | Study design | Mean/ age range (year) | ART Status | No of cases | P | 95% CI | NOS |
|---|---|---|---|---|---|---|---|---|---|---|---|
| 1 | Adal M, Howe R, Kassa D et al | 2018 | Ethiopia | 594 | Cross-sectional | 34 | Pre-ART | 87 | 15.1 | 12.2–17.9 | 6 |
| 2 | Akilimali PZ, Musumari PM, Kashala-Abotnes E et al | 2016 | DRC | 583 | Cross-sectional | 41 | On ART | 141 | 24.1 | 20.6–27.6 | 8 |
| 3 | Amza L, Demissie T, Halala Y. | 2017 | Ethiopia | 519 | Cross-sectional | 18–45+ | On ART | 133 | 26.6 | 22.8–30.4 | 8 |
| 4 | Asnakew M. | 2015 | Ethiopia | 340 | Cross-sectional | 35 | On ART | 103 | 31.2 | 26.3–30.1 | 7 |
| 5 | Benzekri NA, Sambou J, Diaw B, | 2015 | Senegal | 109 | Cross-sectional | 45 | On ART | 25 | 22.9 | 15.0–30.8 | 6 |
| 6 | Birhane M, Loha E, Alemayehu FR. | 2015 | Ethiopia | 389 | Cross-sectional | 40 | On ART | 60 | 25 | 20.7–29.3 | 8 |
| 7 | Daka DW, Ergiba MS. | 2020 | Ethiopia | 1062 | Cross-sectional | 16–50+ | On ART | 357 | 34 | 31.1–36.8 | 8 |
| 8 | Daniel M, Mazengia F, Birhanu D. | 2013 | Ethiopia | 408 | Cross-sectional | 18–45+ | Pre & on ART | 104 | 25.5 | 21.3–29.7 | 8 |
| 9 | M Dedha, M Damena, G Egata et al | 2017 | Ethiopia | 459 | Cross-sectional | 35 | On ART | 131 | 30 | 25.8–34.2 | 7 |
| 10 | Fentie M, Wassie MM, Tesfahun A, | 2017 | Ethiopia | 317 | Cross-sectional | 39 | On ART | 57 | 18.3 | 14.0–22.6 | 6 |
| 11 | Fufa H, Umeta M, Taffesse S et al | 2009 | Ethiopia | 153 | Cross-sectional | 27 | Pre-ART | 27 | 18 | 11.9–24.1 | 7 |
| 12 | Gebru TH, Mekonen HH, Kiros KG. | 2020 | Ethiopia | 394 | Cross-sectional | 41 | On ART | 169 | 42.9 | 38.0–47.8 | 7 |
| 13 | Gedle D, Gelaw B, Muluye D, | 2015 | Ethiopia | 305 | Cross-sectional | 40 | On ART | 77 | 25.2 | 20.3–30.1 | 8 |
| 14 | Girma M, Motuma A, Negasa L. | 2017 | Ethiopia | 502 | Cross-sectional | 24 | Pre & on ART | 133 | 26.5 | 22.6–30.4 | 8 |
| 15 | Gebremichael DY, Hadush KT, Kebede EM, et al | 2018 | Ethiopia | 512 | Cross-sectional | 34 | On ART | 119 | 23.6 | 19.9–27.3 | 8 |
| 16 | Hadgu TH, Worku W, Tetemke D, et al | 2013 | Ethiopia | 276 | Cross-sectional | 33 | On ART | 159 | 42.3 | 36.5–48.1 | 6 |
| 17 | Hailemariam S, Bune GT, Ayele HT et al | 2013 | Ethiopia | 520 | Cross-sectional | 34 | On ART | 64 | 12.3 | 9.4–15.1 | 8 |
| 18 | Kabalimu TK, Sungwa E, Lwabukuna WC. | 2018 | Tanzania | 125 | Cross-sectional | 20–60+ | On ART | 24 | 19.4 | 12.5–26.3 | 6 |
| 19 | Kenea MA, Garoma S, Gemede HF. | 2015 | Ethiopia | 423 | Cross-sectional | 15–50+ | On ART | 112 | 26.47 | 22.3–30.7 | 7 |
| 20 | Getaw Kume | 2017 | Ethiopia | 457 | Cross-sectional | 41 | On ART | 53 | 12.3 | 9.3–15.3 | 8 |
| 21 | Mahlangu K, Modjadji P, Madiba S. | 2020 | South Africa | 480 | Cross-sectional | 35 | On ART | 62 | 13 | 9.9–16.0 | 7 |
| 22 | Mitiku A, Ayele TA, Assefa M, et al | 2016 | Ethiopia | 452 | Cross-sectional | 35 | On ART | 105 | 23.2 | 19.3–27.1 | 7 |
| 23 | Motuma A, Abdeta T. | 2021 | Ethiopia | 502 | Cross-sectional | 37 | On ART | 133 | 26.5 | 22.6–30.4 | 8 |
| 24 | Mulu H, Hamza L, Alemseged F. | 2016 | Ethiopia | 109 | Cross-sectional | 33 | On ART | 51 | 46.8 | 37.4–56.2 | 6 |
| 25 | Naidoo K, Yende-Zuma N, Augustine S | 2018 | South Africa | 1000 | Retrospective cohort | NR | On ART | 149 | 15.7 | 13.5–17.9 | 7 |
| 26 | Nanewortor BM, Saah FI, Appiah PK, et al | 2021 | Ghana | 152 | Cross-sectional | 39 | On ART | 21 | 13.8 | 8.3–19.3 | 6 |
| 27 | Nigusso FT, Mavhandu-Mudzusi AH | 2020 | Ethiopia | 390 | Cross-sectional | 36 | On ART | 232 | 60 | 55.1–64.9 | 7 |
| 28 | Nnyepi MB | 2009 | Botswana | 145 | Cross-sectional | 33 | Pre & on ART | 41 | 28.5 | 21.2–35.9 | 6 |
| 29 | Odwee A, Kasozi KI, Acup CA, | 2020 | Uganda | 253 | Cross-sectional | 39 | On ART | 26 | 10.28 | 6.5–14.0 | 7 |
| 30 | Oumer B, Boti N, Hussen S, et al | 2019 | Ethiopia | 333 | Cross-sectional | 33 | On ART | 79 | 23.72 | 19.2–28.3 | 7 |
| 31 | Regassa TM, Gudeta TA. | 2020 | Ethiopia | 1007 | Cross-sectional | 18–50+ | On ART | 154 | 16 | 13.7–18.3 | 8 |
| 32 | Sahile AT, Ayehu SM, Fanta SF. | 2021 | Ethiopia | 319 | Cross-sectional | 31–41+ | On ART | 61 | 19.1 | 14.8–23.4 | 6 |
| 33 | Saito A, Karama M, Kamiya Y. | 2020 | Kenya | 251 | Cross-sectional | 38 | On ART | 21 | 8.3 | 4.9–11.7 | 6 |

*(Continued)*

**Table 2.** (Continued)

| S.N | Authors | Publication Year | Country | Sample Size | Study design | Mean/ age range (year) | ART Status | No of cases | P | 95% CI | NOS |
|---|---|---|---|---|---|---|---|---|---|---|---|
| 34 | Saliya MS, Azale T, Alamirew A, | 2018 | Ethiopia | 428 | Cross-sectional | 36 | On ART | 97 | 24.1 | 20.1–28.2 | 7 |
| 35 | Shifera N, Molla A, Mesafint G, et al | 2020 | Ethiopia | 402 | Cross-sectional | 34 | On ART | 115 | 29.2 | 24.8–33.6 | 7 |
| 36 | Sunguya BF, Ulenga NK, Siril H, et al | 2017 | Tanzania | 3993 | Cross-sectional | 38 | Pre-ART | 1106 | 27.7 | 26.3–29.1 | 9 |
| 37 | Takarinda KC, Mutasa-Apollo T, Madzima B, et al | 2017 | Zimbabwe | 1,242 | Cross-sectional | 41 | On ART | 122 | 10 | 8.3–11.7 | 8 |
| 38 | Takele AE, Engida AR. | 2017 | Ethiopia | 295 | Cross-sectional | 34 | On ART | 71 | 24 | 19.1–28.9 | 7 |
| 39 | Teklu T, Chauhan NM, Lemessa F, et al | 2020 | Ethiopia | 519 | Cross-sectional | 41 | On ART | 95 | 18.3 | 14.9–21.6 | 7 |
| 40 | Teshome | 2017 | Ethiopia | 302 | Cross-sectional | 36 | On ART | 81 | 27.2 | 22.2–32.2 | 6 |
| 41 | Wasie B, Kebede Y, Yibre A. | 2010 | Ethiopia | 331 | Cross-sectional | 34 | On ART | 92 | 27.8 | 22.9–32.6 | 6 |
| 42 | Wasihun Y, Yayehrad M, Dagne S et al | 2020 | Ethiopia | 350 | Cross-sectional | 35 | On ART | 85 | 26.9 | 22.3–31.6 | 6 |
| 43 | Yitbarek GY, Engidaw MT, Ayele BA, et al | 2020 | Ethiopia | 263 | Cross-sectional | 38 | On ART | 30 | 11.9 | 7.9–15.8 | 6 |
| 44 | Zemede Z, Tariku B, Kote M, et al | 2019 | Ethiopia | 351 | Cross-sectional | 40 | On ART | 64 | 18.23 | 14.2–22.3 | 7 |

DRC: Democratic Republic of Kongo

ART: Antiretroviral therapy, NR: Not reported

**Table 3. Sub-group analysis of the prevalence of undernutrition among adults living with HIV in sub-Saharan Africa by country, study design, and publication year, 2009–2021.**

| Variables | Responses | No of studies | Pooled prevalence (95% CI) | $I^2$ (p-value) |
|---|---|---|---|---|
| Country | Ethiopia | 33 | 25.8% (22.4–29.3) | 95.9% (0.001) |
| | Tanzania | 2 | 24.3% (16.3–32.3) | 81.08% (0.022) |
| | South Africa | 2 | 14.5% (14.9–17.2) | 49.33% (0.16) |
| | Kenya | 1 | 8.3% (4.9–11.7) | - |
| | DRC | 1 | 24.1% (20.6–27.6) | - |
| | Senegal | 1 | 22.9% (15.0–30.8) | - |
| | Botswana | 1 | 28.5% (21.2–35.9) | - |
| | Ghana | 1 | 13.8% (8.3–19.3) | - |
| | Uganda | 1 | 10.3% (6.5–14.0) | - |
| | Zimbabwe | 1 | 10% (8.3–11.6) | - |
| Study design | Cross-sectional | 43 | 23.9% (20.9–26.9) | 96.67% (0.001) |
| | Retrospective cohort | 1 | 15.7% (13.4–17.9) | - |
| Publication year | 2009–2015 | 11 | 25.8% (21.3–30.3) | 89.72% (0.001) |
| | 2016–2021 | 33 | 23.1% (19.4–26.8) | 97.52% (0.001) |
| ART characteristics | Pre-ART | 3 | 20.5% (12.6–28.3) | 95.32% (0.001) |
| | Both "on & pre" ART | 3 | 26.4% (23.7–29.1) | 0.00% (0.783) |
| | On ART | 38 | 23.8 (20.4–27.2) | 96.84% (0.001) |

NB: meta-analysis works when two or more effect estimates are reported

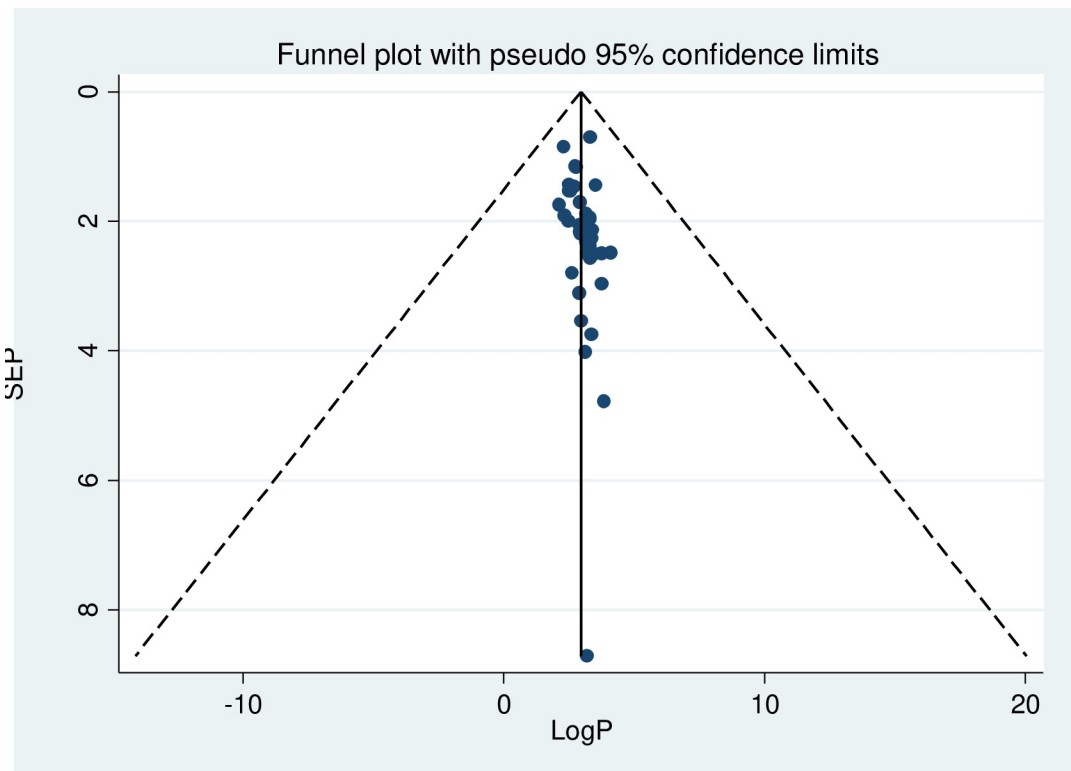

**Fig 2. Funnel plot of studies on undernutrition among adult PLWHIV in SSA, 2009–2021.**

source of high I2 is not identified, though it is expected to rise in a meta-analysis of proportions in different countries, and the result should be interpreted conservatively [81]. Furthermore, the study also demonstrated a wide prediction interval, a direct and easily interpretable indicator as compared to the confidence interval, implying evidence of high heterogeneity. Regarding the publication bias, Egger's regression test (B1 = 5.72, p = 0.002) showed there was publication bias but studies looked relatively symmetrical in the funnel plot (**Fig 2**). However, these indicators of publication bias were developed in the context of comparative data and may not be reliable indicators of publication bias in a meta-analysis of proportions [82].

Sub-group analysis and trim and fill analysis were also performed to deal with the publication bias and heterogeneity S3 Table. A sensitivity analysis was performed and all estimates were within the confidence interval limit, and no individual study contributed to the publication bias. Consequently, it is unnecessary to exclude studies from the final meta-analysis.

## Meta-analysis results

**Prevalence of undernutrition among adult PLWHIV in sub-Saharan Africa.** Of the 44 studies that reported a proportion of undernutrition, the highest prevalence (60%) was reported in Ethiopia [58], whereas the lowest (8.3%) was reported in Kenya [71]. Majority of the studies are reported from Ethiopia (33 studies), cross-sectional design (43 studies), carried out during 2016–2021 (33 studies), and on patients taking ART (38 studies).

In this study, the pooled prevalence of undernutrition using a random-effect model meta-analysis was found to be 23.74% (95% CI: 20.77–26.73) (**Fig 3**). Sub-group analysis by country showed 25.8% (95% CI: 22.4–29.3) in Ethiopia, 14.5% (95% CI: 14.9–17.2) in South Africa, and

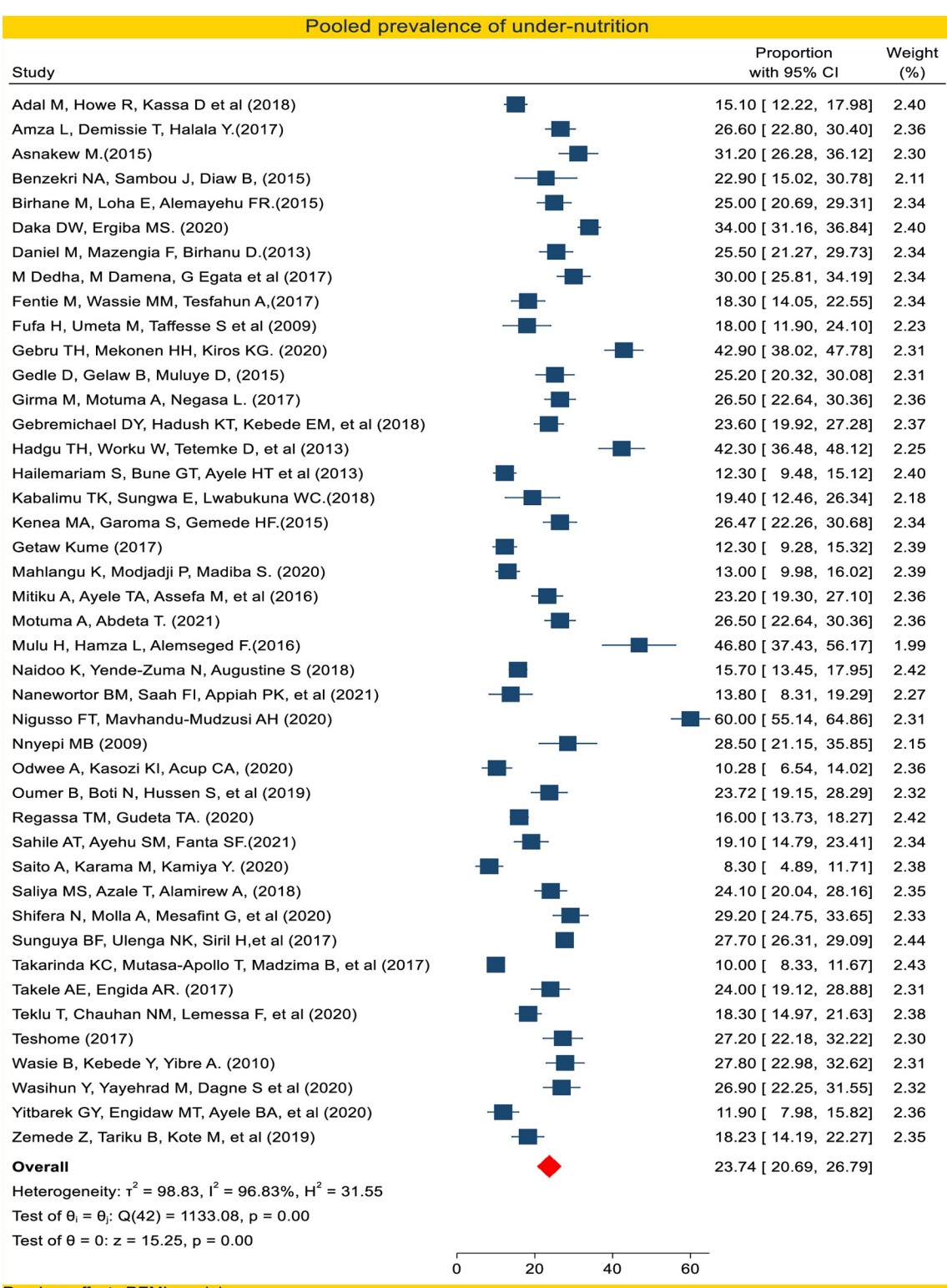

**Fig 3. Forest plot of pooled prevalence of undernutrition among adult PLWHIV in sub-Saharan Africa 2021.**

**Table 4. Summary of the factors associated with undernutrition among adults living with HIV in sub-Saharan Africa, 2021.**

| Factor | No of included studies | Pooled AOR (95% CI) | $I^2$ (p-value) | Reference category |
|---|---|---|---|---|
| WHO clinical stage | 16 | 3.25 (2.57–3.94) | $I^2$ = 0.0% (0.233) | WHO clinical Stage I |
| CD4 count | 10 | 1.91 (1.53–2.29) | $I^2$ = 0.0% (0.535) | CD4 > 500 cells /mm$^3$ |
| Age | 4 | 0.51 (0.31–0.71) | $I^2$ = 31.2% (0.11) | 19–30 years |
| Sex | 4 | 2.11 (1.52–2.7) | $I^2$ = 0.0% (0.534) | Females |
| ART duration | 4 | 2.31 (1.6–3.02) | $I^2$ = 0.0% (0.001) | ≥12 months |

24.3% (95% CI: 16.3–32.3) in Tanzania. Additional sub-group analysis by ART status, by study design, and by publication year also carried out (**Table 3**).

**Factors associated with undernutrition among adults of PLWHIV in sub-Saharan Africa.** From the searched published articles 16 reported "WHO clinical stage" [13, 39, 41–43, 45, 47, 50, 52, 57, 59, 60, 64–66, 68, 83], four studies reported "CD4 count" [40, 43, 47, 55, 57, 59, 60, 80, 83], four studies reported patient's "age" [13, 39, 55, 56, 60, 77], four studies reported "sex" [40, 56, 64, 80, 83], and four articles reported "duration of ART treatment" [8, 59, 64, 67, 68] as factors associated with undernutrition among adult PLWHIV in sub-Saharan Africa (**Table 4**). Patients living with HIV and WHO clinical stage III/IV were 3.25 (AOR, 95% CI: 2.57–3.93) times higher odds of developing undernutrition as compared to WHO clinical stage I/II (**Fig 4**). Similarly, patients whose CD4 count was less than 200cells/mm$^3$ were 1.94 times (AOR = 1.94, 95% CI: 1.53–2.28) with higher odds of developing undernutrition as compared to their counterparts (CD4 >500cells/mm$^3$) (**Fig 5**).

Regarding the age of study participants, patients aged 40 years and above had 49% lower odds of developing undernutrition as compared to those aged 19 to 30 years (AOR = 0.51, 95% CI: 0.26–0.76) (**Fig 6**). Furthermore, the odds of developing undernutrition among males living with HIV were 2 times (AOR = 2.11, 95% CI: 1.52–2.7) higher as compared to female patients (**Fig 7**). Similar to this, patients receiving ART for less than 12 months had 2.68 times the risk of developing undernutrition compared to individuals taking it for more than 12 months (**Fig 8**). This implies that the longer the duration of patients' taking ART, the lower the risk of developing under-nutrition.

## Discussion

Despite the improvement of comprehensive HIV care, sub-Saharan Africa continues to be an epicenter of HIV transmission and has a high prevalence of malnutrition among adults living with HIV. This study aimed to investigate the pooled prevalence and correlates of undernutrition among adults living with HIV/AIDS. The result shows a significant number of adults living with HIV are malnourished, and several socio-demographic and clinical factors have been associated with undernutrition.

In this meta-analysis, the pooled prevalence of undernutrition (adult BMI <18.5kg/m$^2$) among adults living with HIV was high (23.74%) and interpreted as a serious situation according to WHO nutrition landscape information system cut-off values (20–39%) [84]. It is serious because studies have shown that undernutrition increases the risk of opportunistic infections (OIs) and mortality [4, 85]. The pooled prevalence slightly decreased from 25.8% in 2009–2015 to 23% in 2015–2021. This might be attributed to the expansion of nutrition intervention programs and the improvement of comprehensive HIV care in Africa. This finding is in line with a previous meta-analysis study in Ethiopia, in which the pooled prevalence was reported as 26% (95% CI: 22–30%) [14].

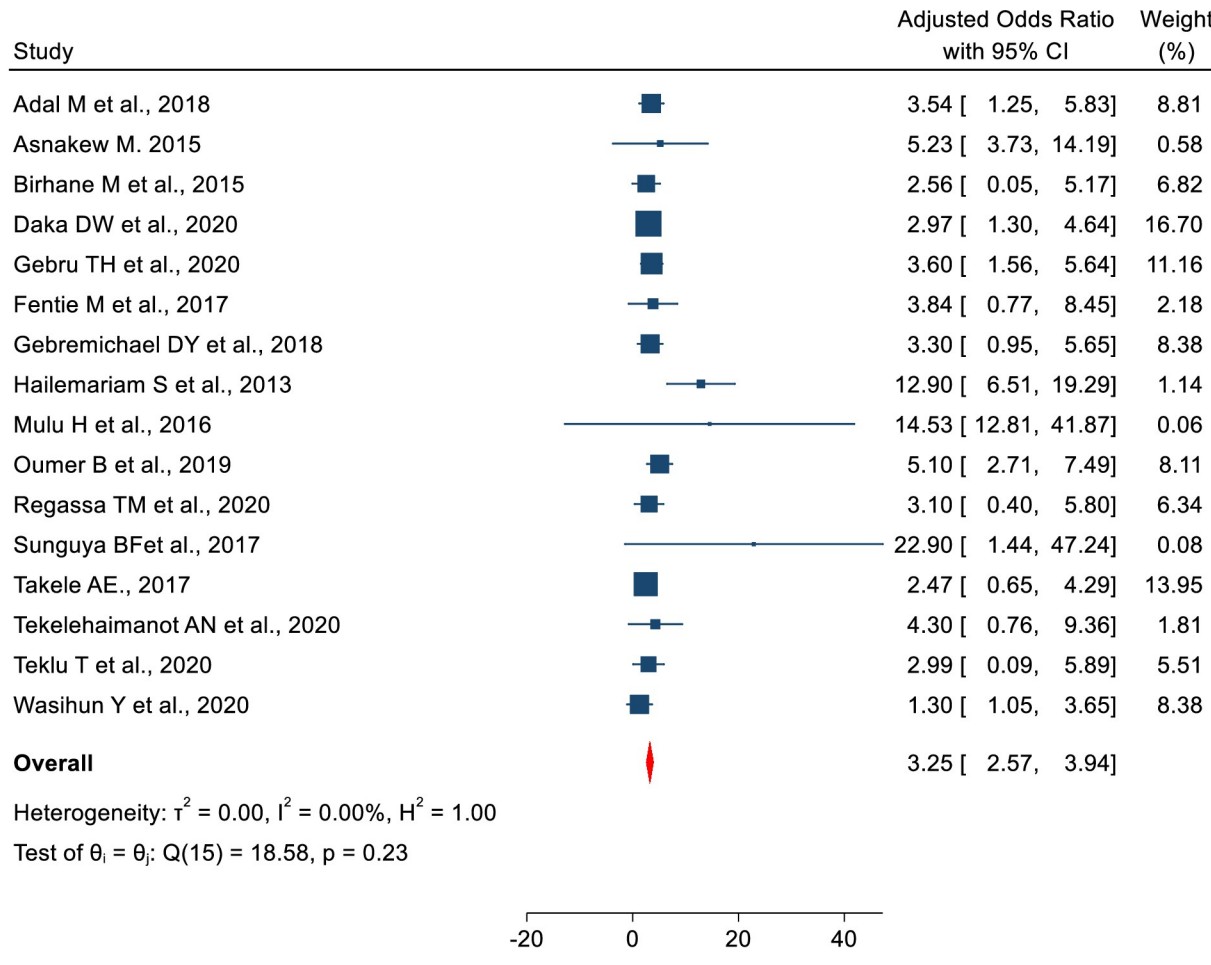

| Study | | Adjusted Odds Ratio with 95% CI | Weight (%) |
|---|---|---|---|
| Adal M et al., 2018 | | 3.54 [ 1.25, 5.83] | 8.81 |
| Asnakew M. 2015 | | 5.23 [ 3.73, 14.19] | 0.58 |
| Birhane M et al., 2015 | | 2.56 [ 0.05, 5.17] | 6.82 |
| Daka DW et al., 2020 | | 2.97 [ 1.30, 4.64] | 16.70 |
| Gebru TH et al., 2020 | | 3.60 [ 1.56, 5.64] | 11.16 |
| Fentie M et al., 2017 | | 3.84 [ 0.77, 8.45] | 2.18 |
| Gebremichael DY et al., 2018 | | 3.30 [ 0.95, 5.65] | 8.38 |
| Hailemariam S et al., 2013 | | 12.90 [ 6.51, 19.29] | 1.14 |
| Mulu H et al., 2016 | | 14.53 [ 12.81, 41.87] | 0.06 |
| Oumer B et al., 2019 | | 5.10 [ 2.71, 7.49] | 8.11 |
| Regassa TM et al., 2020 | | 3.10 [ 0.40, 5.80] | 6.34 |
| Sunguya BFet al., 2017 | | 22.90 [ 1.44, 47.24] | 0.08 |
| Takele AE., 2017 | | 2.47 [ 0.65, 4.29] | 13.95 |
| Tekelehaimanot AN et al., 2020 | | 4.30 [ 0.76, 9.36] | 1.81 |
| Teklu T et al., 2020 | | 2.99 [ 0.09, 5.89] | 5.51 |
| Wasihun Y et al., 2020 | | 1.30 [ 1.05, 3.65] | 8.38 |
| **Overall** | | 3.25 [ 2.57, 3.94] | |

Heterogeneity: $\tau^2 = 0.00$, $I^2 = 0.00\%$, $H^2 = 1.00$

Test of $\theta_i = \theta_j$: Q(15) = 18.58, p = 0.23

Random-effects REML model

**Fig 4. The pooled effect of "WHO clinical stage" on undernutrition among adult PLWHIV in sub-Saharan Africa, 2021.**

In contrast, our finding is higher as compared to a previous meta-analysis study conducted on women living with HIV in sub-Saharan Africa: 10.3% (95% CI: 7.4%–14.1%). The study used secondary data from the DHS and analyzed the reported estimate of only 11 sub-Saharan African countries. The disparity might be explained by the fact that over the last 13 years, there have been changes in socio-demography, the trend of HIV incidence, food insecurity, and other population factors that might have been associated with undernutrition. Moreover, the study was carried out only among women, which may have affected the result. Nevertheless, our study also asserts that men living with HIV are at a higher risk of developing malnutrition as compared to women [37], even though the biological mechanism is not clear.

We also found that patients with an advanced WHO clinical stage, a lower CD4 count, being of male sex, a younger age, and a shorter duration of ART treatment had a higher likelihood of developing undernutrition in adults living with HIV. The factor regarding the WHO clinical stage was also reported in a meta-analysis report in Ethiopia [86]. This may be due to the advanced WHO clinical stage and low CD4 count, which are indicators of severe immune deficiency are directly linked to undernutrition, especially protein and energy malnutrition. Thus, it is important to give due emphasis to nutrition counseling and supplementation with high-energy and protein foods during the follow-up visit.

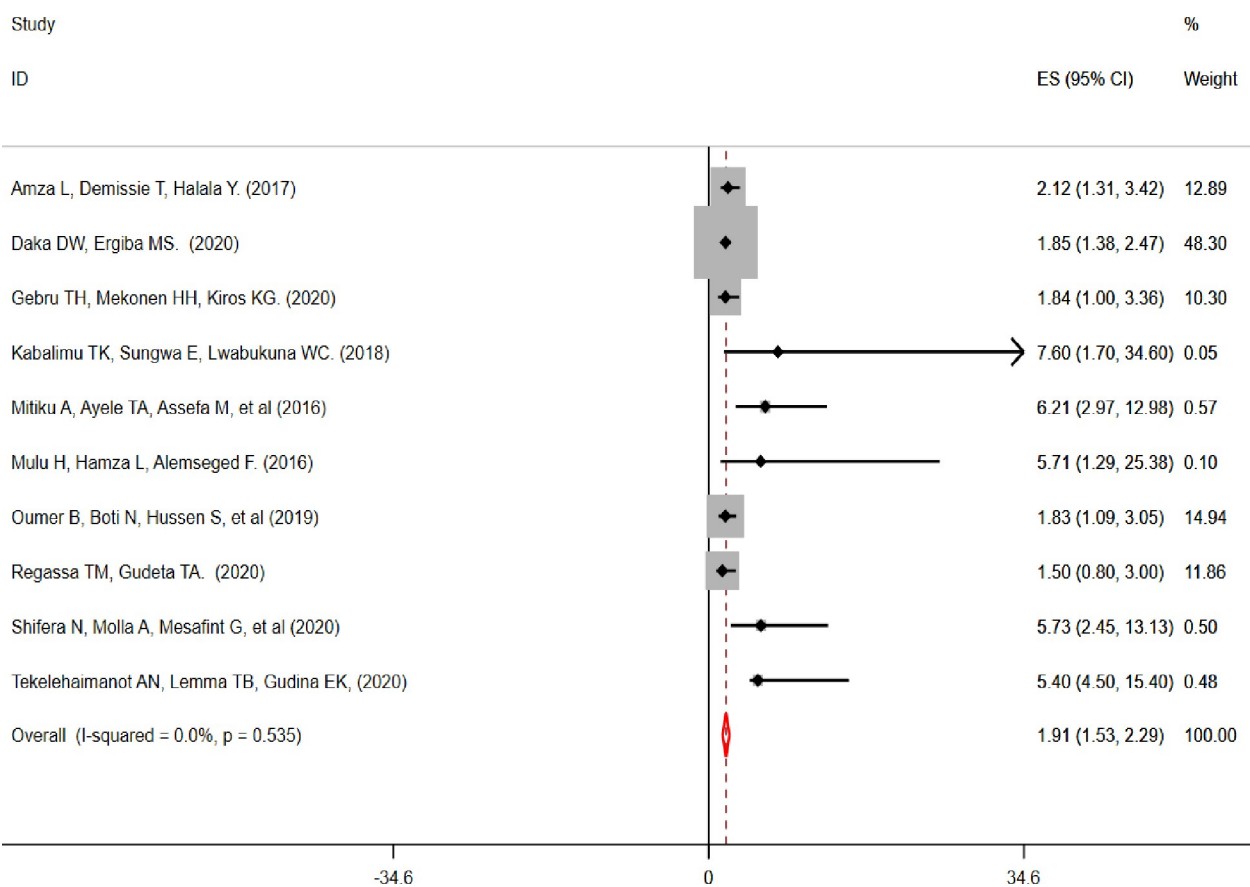

**Fig 5. Pooled effect of "CD4 count" on undernutrition among adult PLWHIV in sub-Saharan Africa, 2021.**

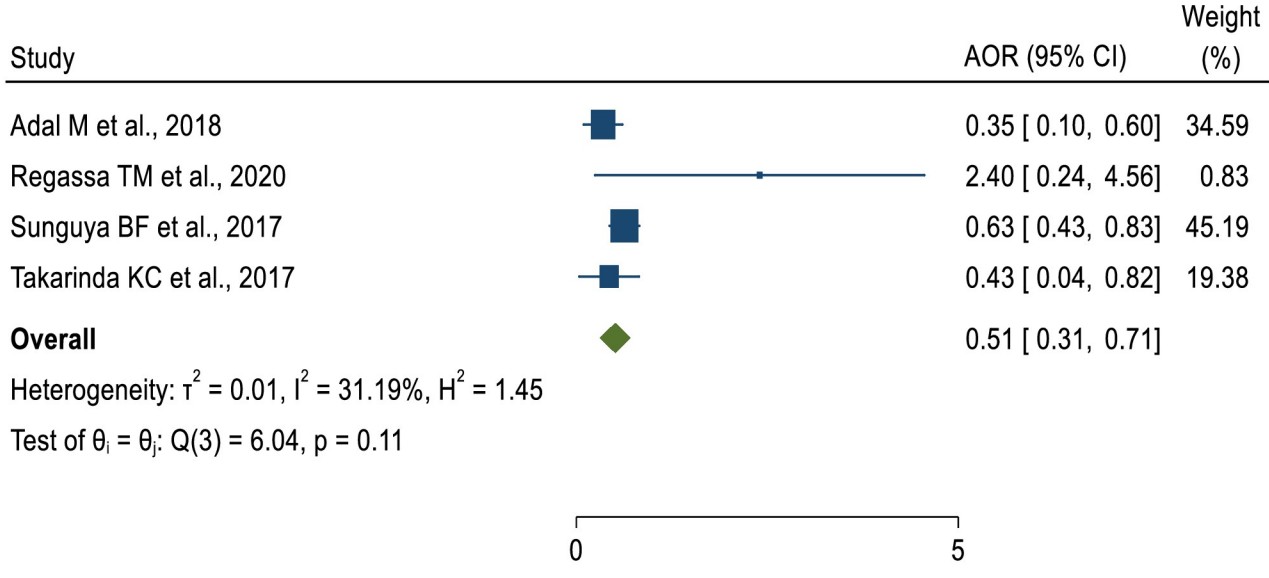

**Fig 6. Pooled effect of "age" of study participants on undernutrition among adult PLWHIV in sub-Saharan Africa, 2021.**

| Study | | Effect size with 95% CI | Weight (%) |
|---|---|---|---|
| Amza L et al., 2017 | | 1.80 [ 0.88, 2.72] | 40.99 |
| Kabalimu TK et al., 2018 | | 5.10 [ 1.41, 11.61] | 0.82 |
| Motuma A et al., 2021 | | 2.11 [ 1.23, 2.99] | 44.72 |
| Tekelehaimanot AN et al., 2020 | | 2.90 [ 1.29, 4.51] | 13.47 |
| **Overall** | | 2.11 [ 1.52, 2.70] | |

Heterogeneity: $\tau^2 = 0.00$, $I^2 = 0.00\%$, $H^2 = 1.00$

Test of $\theta_i = \theta_j$: Q(3) = 2.17, p = 0.54

Random-effects REML model

**Fig 7. Pooled effect of "sex" on undernutrition among adult PLWHIV in sub-Saharan Africa, 202.**

Furthermore, young adults and ART beginners were identified as being at risk for malnutrition. The reason for the younger ages might be due to the poor emotional readiness to accept the disease condition and the failure to receive comprehensive HIV care at an early age. On the other hand, increasing age may improve acceptance and the perceived benefits of adherence to recommendations by health care providers. However, the result for younger ages requires further exploration. In spite of this, although it is acknowledged that receiving ART

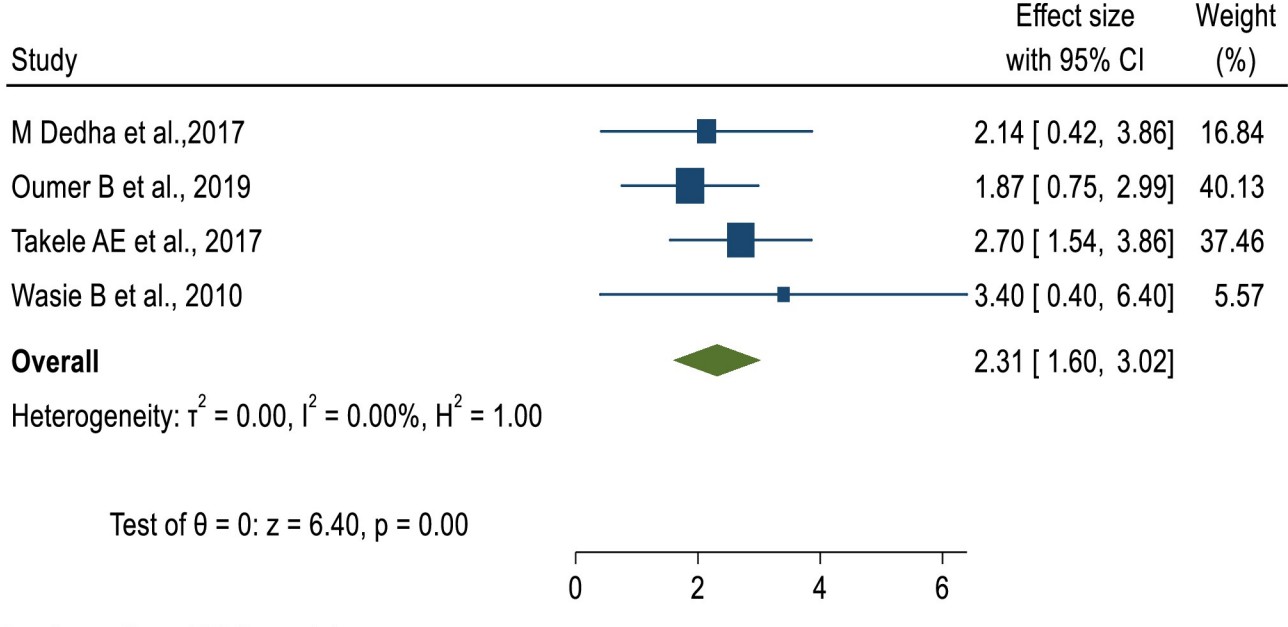

| Study | | Effect size with 95% CI | Weight (%) |
|---|---|---|---|
| M Dedha et al.,2017 | | 2.14 [ 0.42, 3.86] | 16.84 |
| Oumer B et al., 2019 | | 1.87 [ 0.75, 2.99] | 40.13 |
| Takele AE et al., 2017 | | 2.70 [ 1.54, 3.86] | 37.46 |
| Wasie B et al., 2010 | | 3.40 [ 0.40, 6.40] | 5.57 |
| **Overall** | | 2.31 [ 1.60, 3.02] | |

Heterogeneity: $\tau^2 = 0.00$, $I^2 = 0.00\%$, $H^2 = 1.00$

Test of $\theta = 0$: z = 6.40, p = 0.00

Random-effects REML model

**Fig 8. Pooled effect of "ART duration" on undernutrition among adult PLWHIV in sub-Saharan Africa, 2021.**

improves nutritional status, the impact of HIV on nutrition begins even before diagnosis and needs a longer course of therapy in order to be reversed, and noticed by anthropometric measurements.

## Limitation of the study

The possible limitation for this review was the inability of accessing some databases like EMBASE, CINHAL, and Scopus. This was compensated by searching for published articles in broad databases like Google scholar. The other limitation is the absence of similar meta-analysis studies for comparison of our result. There is an uneven distribution of included studies among countries, a large number of which were reported from Ethiopia. Additionally, there is high heterogeneity, and studies published only in English were included.

## Conclusion

The pooled prevalence of undernutrition among adult PLWHIV in sub-Saharan Africa remained high. WHO clinical stage, CD4 count, duration of ART treatment, age, and sex were found to be the factors associated with undernutrition. Reinforcing nutrition counseling, care, and support for adults living with HIV is recommended. Priority nutritional screening and interventions should be provided for patients with advanced WHO clinical stages, low CD4 counts, the male gender, younger age groups, and ART beginners.

## Supporting information

**S1 Table. Data extraction sheet used in the meta-analysis of the prevalence and associated factors of undernutrition among adults taking antiretroviral therapy in sub-Saharan Africa, 2009–2021.**
(XLSX)

**S2 Table. PRISMA statement presentation of systematic review and meta-analysis of undernutrition among adults taking ART in SSA, 2009–2021.**
(DOCX)

**S3 Table. Trim and fill analysis of prevalence of undernutrition and associated factors among adult PLWHIV in SSA, 2009–2021.**
(DOCX)

## Author Contributions

**Conceptualization:** Awole Seid.

**Data curation:** Awole Seid, Yinager Workineh.

**Formal analysis:** Yinager Workineh, Getenet Dessie, Zebenay Workneh Bitew.

**Methodology:** Awole Seid, Getenet Dessie.

**Software:** Getenet Dessie, Zebenay Workneh Bitew.

**Writing – original draft:** Awole Seid.

**Writing – review & editing:** Awole Seid, Omer Seid, Getenet Dessie, Zebenay Workneh Bitew.

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
