## [Decision Letter · Decision Letter 0]

5 May 2022

PONE-D-21-36754Prevalence and associated factors of under-nutrition among adults taking anti-retroviral therapy in sub-Saharan Africa: A systematic review and meta-analysisPLOS ONE

Dear Dr. Ali,

Thank you for submitting your manuscript to PLOS ONE. After careful consideration, we feel that it has merit but does not fully meet PLOS ONE’s publication criteria as it currently stands. Therefore, we invite you to submit a revised version of the manuscript that addresses the points raised during the review process.

We look forward to receiving your revised manuscript.

Kind regards,

Joel Msafiri Francis, MD, MS, PhD

Academic Editor

PLOS ONE

Journal Requirements:

4. We note that you have referenced (ie. Bewick et al. [5]) which has currently not yet been accepted for publication. Please remove this from your References and amend this to state in the body of your manuscript: (ie “Bewick et al. [Unpublished]”) as detailed online in our guide for authors

Reviewers' comments:

Reviewer's Responses to Questions

**Comments to the Author**

1. Is the manuscript technically sound, and do the data support the conclusions?

Reviewer #1: Yes

Reviewer #2: Partly

2. Has the statistical analysis been performed appropriately and rigorously? 

Reviewer #1: Yes

Reviewer #2: Yes

3. Have the authors made all data underlying the findings in their manuscript fully available?

Reviewer #1: Yes

Reviewer #2: Yes

4. Is the manuscript presented in an intelligible fashion and written in standard English?

Reviewer #1: No

Reviewer #2: No

5. Review Comments to the Author

Reviewer #1: Prevalence and associated factors of under-nutrition among adults taking anti-retroviral therapy in sub-Saharan Africa: A systematic review and meta-analysis

1. General comment: Relevant topic, consolidating evidence about the relationship between under-nutrition and ART.

However, the whole article needs to be revisited to make sure that English language is well written especially writing the article in a past tense. The authors should define abbreviations used for the first time

2. Abstract:

Background: Lacks the research gap and the broad aim/objective of conducting the study. Method: Some sentences written in present tense instead of past tense.

3. Main body:

Introduction: Revisit the first sentence in the first paragraph (Probably the authors meant that the rate of new HIV infection is decreasing Worldwide). 5th and 6th paragraphs talked about previous studies which also determined prevalence of under-nutrition and associated factors with HIV. However, the authors did not show the gap intended to be filled by the current study. It is not clear why the current meta-analysis was done as there were already some meta-analysis done as indicated in the literature review. However, the rationale is well stated.

Method: Include a table showing the PICO of the systematic review. Search words shown are only those for PubMed. I suggest you list all the search words considered in every search engine without giving the details of how the different search words were related (OR/AND). Give initials for the third author.

Results: Arrange the table and figures e.g. the PRISMA as they are discussed in the results section.

Discussion: Avoid repeating results in the discussion section

Reviewer #2: This was a systematic review to estimate pooled prevalence of under nutrition and associated factors among PLHIV in sSA.

Comment:

Overall:

Manuscript need English proof reading. Several areas past tense have not been used, conjunction missing. Message not clearly communicated

Title: Consider revising it to “Prevalence of under-nutrition and its associated factors among adults taking anti-retroviral therapy in sub-Saharan Africa: A systematic review and meta-analysis”

Define HIV appropriately as Human immunodeficiency virus. Do dot prefer people as HIV infected, you may consider using people living with HIV as the former is regarded as stigmatizing

Introduction:

There are about 7 paragraphs. Not clearly aligned. Consider reducing number of paragraphs to about 4 and each should carry specific message. Stating bidirectional relationship between ART and undernutrition without describing it not enough.

22 times more likely, if the study did not use risk ration, interpretation is not right.

Methods:

Needs English proof reading

Issues such as ‘The review is aimed……’; ‘databases are accessed…..’ needs revision to past tense. The word on the other had was used severally in the manuscript but not showing clear contrast

‘…..to find associated factors of under-nutrition’ could be factors associated with under-nutrition

Results.

There is unnecessary bolding of numbers or reference (eg figure xxx) , a total of 22316 etc

‘Sensitivity analysis is performed…..’ Needs to be past tense through out the manuscript

Odds ratios were not appropriately interpreted. The odds of X is 2 times that of Y. Not 2 times higher

‘……1.94 times more likely ….’ Is not appropriate, can be written 1.94 higher odds of having under-nutrition. This is not a risk ratio, you can’t say more likely

Discussion:

Needs revision

Like introduction, consider few paragraph with specific message

1- Paragraph- summarize main findings

2-3 Paragraphs- consider main factors to expand

4- para- consider limitations and strength

5- conclusion

6. PLOS authors have the option to publish the peer review history of their article (what does this mean?). If published, this will include your full peer review and any attached files.

Reviewer #1: No

Reviewer #2: No

---

## [Author Response · Author response to Decision Letter 0]

14 May 2022

Response to reviewer #1

General comment: the whole article needs to be revisited to make sure that English language is well written especially writing the article in a past tense. The authors should define abbreviations used for the first time. The authors should define abbreviations used for the first time.

Response: We noted several English proofreading problems in the manuscript. We have made corrections word by word. I also invited English language editors to minimize language problems. Grammatical errors are also corrected.

Yes abbreviations should be defined in the first use and we made corrections.

Comment on abstract: 

Background: Lacks the research gap and the broad aim/objective of conducting the study. Method: Some sentences written in present tense instead of past tense.

Response: We incorporate the research gap and objective in the background section of the abstract. We changed verbs to past tense in the method section.

Comment 

1. Main body:

Introduction: Revisit the first sentence in the first paragraph (Probably the authors meant that the rate of new HIV infection is decreasing Worldwide). 5th and 6th paragraphs talked about previous studies which also determined prevalence of under-nutrition and associated factors with HIV. However, the authors did not show the gap intended to be filled by the current study. It is not clear why the current meta-analysis was done as there were already some meta-analysis done as indicated in the literature review. However, the rationale is well stated.

Response: The first and second paragraphs are merged and paraphrased. Meta-analysis was done in Ethiopia, not at the sub-Sahara Africa level. There are a number of studies but lacks consistency. Besides, we believe a single country figure will not represent the region. Therefore, this is the first review in sub-Sahara Africa and this statement is incorporated in the paragraph.

Comment 

Method: Include a table showing the PICO of the systematic review. Search words shown are only those for PubMed. I suggest you list all the search words considered in every search engine without giving the details of how the different search words were related (OR/AND). Give initials for the third author. 

Response: Thank you for this comment and PICO table is incorporated and makes the study more clear. Search words and Mesh terms were used only for PubMed because it is an open access database. We have no institutional access to Embase, Scopus and other databases which is described under limitation. Therefore, we didn’t use different search words for these databases. However, we searched articles in Google scholar which is comprehensive and include articles indexed in those databases.

Comment on results: Arrange the table and figures e.g. the PRISMA as they are discussed in the results section.

Response: We arranged and make results in line with the PRISMA chart.

Comment on discussion: Avoid repeating results in the discussion section

Response: We noted some repetition of results in the discussion section and removed it.

Response to reviewer #2

Overall comment:

Manuscript need English proof reading. Several areas past tense have not been used, conjunction missing. Message not clearly communicated

Title: Consider revising it to “Prevalence of under-nutrition and its associated factors among adults taking anti-retroviral therapy in sub-Saharan Africa: A systematic review and meta-analysis”. Define HIV appropriately as Human immunodeficiency virus. Do dot prefer people as HIV infected, you may consider using people living with HIV as the former is regarded as stigmatizing

Response: We admit the manuscript had language problems and we undergo thorough revision and language editors were also invited to help us communicate the message clearly. The comment on title sounds more logical if the title is rephrased and done as indicated by the reviewer. The abbreviation HIV is defined appropriately. We also accept the comment of stigmatizing expression of “HIV infected…” We corrected the expression throughout the manuscript.

Comment 

Introduction:

There are about 7 paragraphs. Not clearly aligned. Consider reducing number of paragraphs to about 4 and each should carry specific message. Stating bidirectional relationship between ART and under-nutrition without describing it not enough. 22 times more likely, if the study did not use risk ration, interpretation is not right.

Response: 

The paragraphs in the introduction are abridged to four and each carries specific messages. Regarding the interpretation of the odds ratio, we agree it is preferable to interpret as indicated and we modified throughout the manuscript.

Comment 

Methods:

Needs English proof reading

There is unnecessary bolding of numbers or reference (e.g. figure xxx) , a total of 22316 etc

Response:

We changed the tenses in methodology and other sections as commented.

Unnecessary bolding of words or numbers are avoided. We tried to correct the grammatical errors made throughout the manuscript including in the result section.

Comments 

Discussion:

Needs revision

Like introduction, consider few paragraph with specific message

1- Paragraph- summarize main findings

2-3 Paragraphs- consider main factors to expand

4 - para- consider limitations and strength

 5 – Conclusion

Response: 

In the discussion section, we also merge paragraphs with similar messages. Even though there are few studies that limit the discussion, we tried to add other interpretation guides.

---

## [Decision Letter · Decision Letter 1]

21 Jun 2022

PONE-D-21-36754R1Prevalence of under-nutrition and associated factors among adults taking antiretroviral therapy in sub-Saharan Africa: A systematic review and meta-analysisPLOS ONE

Dear Dr. Ali,

Thank you for submitting your manuscript to PLOS ONE. After careful consideration, we feel that it has merit but does not fully meet PLOS ONE’s publication criteria as it currently stands. Therefore, we invite you to submit a revised version of the manuscript that addresses the points raised during the review process.

We look forward to receiving your revised manuscript.

Kind regards,

Joel Msafiri Francis, MD, MS, PhD

Academic Editor

PLOS ONE

Reviewers' comments:

Reviewer's Responses to Questions

**Comments to the Author**

1. If the authors have adequately addressed your comments raised in a previous round of review and you feel that this manuscript is now acceptable for publication, you may indicate that here to bypass the “Comments to the Author” section, enter your conflict of interest statement in the “Confidential to Editor” section, and submit your "Accept" recommendation.

Reviewer #1: All comments have been addressed

Reviewer #3: (No Response)

2. Is the manuscript technically sound, and do the data support the conclusions?

Reviewer #1: Yes

Reviewer #3: No

3. Has the statistical analysis been performed appropriately and rigorously? 

Reviewer #1: Yes

Reviewer #3: No

4. Have the authors made all data underlying the findings in their manuscript fully available?

Reviewer #1: Yes

Reviewer #3: Yes

5. Is the manuscript presented in an intelligible fashion and written in standard English?

Reviewer #1: Yes

Reviewer #3: Yes

6. Review Comments to the Author

Reviewer #1: Introduction:

• Third sentence, first paragraph: Look for more than one proof for food insecurity in Africa or better look for a systematic review

• Make sure you define all the abbreviation when used for the first time e.g. ART

• Second paragraph: The bi-directionality was not shown; you only showed that malnutrition worsens HIV. The effect of HIV on malnutrition is missing

Reviewer #3: General comment

The interpretation of this review has a bias towards comparing outcomes for Ethiopia with other published outcomes about Ethiopia. Given that all the authors are from Ethiopia, perhaps the authors want to reframe this study title and objectives to having a particular focus on Ethiopia

Specific comments

Wherever point estimates are provided confidence intervals should be included – for both this study, and the supporting background text

Methods

Inclusion criteria – prospective studies were not included?

How was undernutrition defined?

NewCastle Ottawa is good but it is stated that “A score of greater or equal to 6 points was considered “good” and included in the study”. Summary scores are a misleading way to assess quality.

Further, if this was done, the number of studies excluded because they were not considered “good” should be included in the flow diagram and results section narrative

A funnel plot is noted in the abstract and results, but not in the methods.

Subgroup and sensitivity analyses are described in the results, but not the methods.

12 is not the most appropriate test for heterogeneity when the outcome measure is a proportion. As such, it is unsurprising that 12 is >90%. (Tau2 would have been better)

The search strategy is not highly sensitive (eg “Anti-Retroviral Therapy” is a very limited term) and this should be noted as a limitation. Preferably, standard search terms would have been used (eg see search terms used by Cochrane, or search strings used by other reviews cited by this review, eg ref 10.)

Also, the search strategy has two sets of terms

“Proportion” or “Prevalence” or “Magnitude” AND“Malnutrition” OR “Under-nutrition” OR “Under-weight” OR “Wasting” AND “HIV-positive” OR “HIV-infected” OR “Anti-Retroviral Therapy” AND “Adults” AND “ (each country) in subSaharan Africa”.

AND

“Associated factors” OR “Determinants” OR “Predictors” OR “Correlates” AND “malnutrition” OR “under-nutrition” OR “under-weight” OR “Wasting” AND “HIV-positive” OR “HIV-infected” OR “Anti-Retroviral Therapy” AND “Adults” AND “ (each country) Sub-Saharan Africa”

These two sets of terms have duplicate terms. Limiting the search to studies that contain the words “Proportion” or “Prevalence” or “Magnitude” or “Correlates” is quite a crude limitation, as studies could report these outcomes without using these terms.

Results

“Regarding the publication year, 11 articles were published in 2020 followed by 9 articles in 2017. Besides, only one article was reported in 2010”

This is an anecdotal way of describing year of publication, given that 44 studies were included. Please reconsider how to present these data

Tables need legends to explain the acronyms

Figure 3

This is a very unconventional forest plot that does not display pooled estimates and confidence intervals well

7. PLOS authors have the option to publish the peer review history of their article (what does this mean?). If published, this will include your full peer review and any attached files.

Reviewer #1: No

Reviewer #3: No

---

## [Author Response · Author response to Decision Letter 1]

18 Jul 2022

Response to reviewer #1

Comment: Third sentence, first paragraph: Look for more than one proof for food insecurity in Africa or better look for a systematic review

Response: I have included additional systematic review and primary studies to strengthen the evidence. 

Comment: Make sure you define all the abbreviation when used for the first time e.g. ART

Response: Abbreviations are defined in the first usage throughout the manuscript.

Comment: Second paragraph: The bi-directionality was not shown; you only showed that malnutrition worsens HIV. The effect of HIV on malnutrition is missing.

Response: The effect of HIV on malnutrition is presented in detail.

Response to reviewer # 3

Comment: The interpretation of this review has a bias towards comparing outcomes for Ethiopia with other published outcomes about Ethiopia. Given that all the authors are from Ethiopia, perhaps the authors want to reframe this study title and objectives to having a particular focus on Ethiopia.

Response: The use of systematic reviews and meta-analysis studies reported from Ethiopia for interpretation and comparison was due to a lack of similar or related studies in other countries, not the authors' origins in Ethiopia. We tried our best to search for meta-analysis results conducted elsewhere outside of Africa too. Of course, we added one meta-analysis study conducted on a similar topic but carried out only among women for additional comparison and interpretation. Notably, 11 out of 44 studies were conducted outside of Ethiopia, and as long as studies are systematically searched and no eligible studies were left in the final analysis, the result can be taken as representative of sub-Sahara African region. In spite of that, we mentioned the dominancy of studies from Ethiopia under the “limitation of the study”. 

Comment: Wherever point estimates are provided confidence intervals should be included – for both this study, and the supporting background text.

Response: It is accepted and we included confidence intervals for different point estimates presented in the manuscript.

Comment: Inclusion criteria –prospective studies were not included?

Response: All observational studies reporting prevalence, incidence, and/or determinants were searched. But we didn’t find studies with prospective cohort designs, and only one retrospective cohort study was included. 

Comment: How was under-nutrition defined?

Response: Even though there is no single indicator of nutritional status in adults, a low body mass index (BMI) of < 18.5 kg/m2 or underweight is an indicator of advanced malnutrition. The risk becomes higher when patients have chronic diseases like HIV. If it is not prevented or early detected, it will lead to "wasting syndrome," a sign of an advanced immune-compromised state. It was also demonstrated that starting antiretroviral therapy (ART) accelerated mortality (6 months) if the patient's BMI was less than 18. An additional point regarding definition of under-nutrition is added in the manuscript. All of the studies used a BMI of < 18.5kg/m2 to define under-nutrition.

(Koethe, John R., and Douglas C. Heimburger. "Nutritional aspects of HIV-associated wasting in sub-Saharan Africa." The American journal of clinical nutrition 91.4 (2010): 1138S-1142S.

https://emedicine.medscape.com/article/2058483-overview

Comment: Newcastle Ottawa Scale is good but it is stated that “A score of greater or equal to 6 points was considered “good” and included in the study”. Summary scores are a misleading way to assess quality. Further, if this was done, the number of studies excluded because they were not considered “good” should be included in the flow diagram and results section narrative

Response: The Newcastle Ottawa Scale, JBI, and Cochrane risk of bias assessment are the commonly used tools to screen the quality of published studies. When using NOS, it is difficult to make a qualitative judgment for eligibility without a cut-off point. The domain of assessment uses the star method, which is rated out of 10 points for cross-sectional studies. There are also studies published in PLOS ONE using similar judgment. We agree that studies that were not considered "good" should be reported in the flowchart. However, no study is excluded from the analysis due to poor quality. 

Comment: A funnel plot is noted in the abstract and results, but not in the methods.

Response: It is incorporated in the method section as indicated.

Comment: Subgroup and sensitivity analyses are described in the results, but not the methods.

Response: It is incorporated in method section too.

Comment: I2 is not the most appropriate test for heterogeneity when the outcome measure is a proportion. As such, it is unsurprising that I2 is >90%. (Tau2 would have been better).

Yes it is true that both Q test with p value and I2 are a relative measure of heterogeneity. They don’t tell us exactly the magnitude of variation of effect sizes among the studies. Therefore, high I2 in the context of proportional meta-analysis does not necessarily mean that data is inconsistent. As such, the results of this test should be interpreted conservatively. It is advised to use I2 as a criterion for a decision whether a subgroup analysis or moderato analysis is indicated.

On the other hand, τ2 or Tau2 is an estimate of the between-study variance in a random-effects meta-analysis. The square root of this number (i.e. tau) is the estimated standard deviation of underlying effects across studies. T2 is not used itself as a measure of heterogeneity but is used in two other ways: (1) it is used to compute Tau; and (2) it is used to assign weights to the studies in the meta-analysis under the random-effects model. Tau is used for computing the prediction interval. Tau is a useful first indication of the extent of this dispersion. However, the prediction interval is a more direct and more easily interpretable indicator.

Michael Borenstein et al. (2009), Introduction to Meta-Analysis, Chichester (UK): Wiley.

Therefore, in our meta-analysis study the following is incorporated.

Tau2 = 98.83, Tau = 9.94, prediction interval (13.8 – 33.68), 95% CI confidence interval, (20.69 – 26.79)

With high heterogeneity, prediction intervals will be wider than confidence intervals, and can be considered a more conservative way to incorporate uncertainty in the analysis. Where possible, it is suggested estimation of prediction intervals alongside with confidence intervals, especially for prevalence and incidence estimates and we include this concept.

Barker, Timothy Hugh, et al. "Conducting proportional meta-analysis in different types of systematic reviews: a guide for synthesis of evidence." BMC Medical Research Methodology 21.1 (2021): 1-9.

Comment: The search strategy is not highly sensitive (e.g. “Anti-Retroviral Therapy” is a very limited term) and this should be noted as a limitation. Preferably, standard search terms would have been used (e.g. see search terms used by Cochrane, or search strings used by other reviews cited by this review, e.g. ref 10).

Response: We believe that we have searched enough articles using sufficient key words. We were having difficulty finding alternative search terms for the key word "Anti-Retroviral Therapy". Even the suggested reference by the reviewer didn’t include this word. We accept that truncations were not used. Additionally, the Cochrane handbook for systematic review recommends to avoid using too many different search concepts but a wide variety of search terms. (www.training.cochrane.org/handbook). However, we don’t mean that the recommendation for maximizing the number of alternative key words is not important in our future work. Regarding duplication of terms, the second search strategy connected by “AND” was used to access potential case-control and retrospective cohort studies that might only report the associated factors (determinants) without the prevalence or incidence of under-nutrition. We believe that no study is leftover other than based on eligibility criteria in our final meta-analysis and it may not be necessary to put the search strategy as a crude limitation.

Comment: Results “Regarding the publication year, 11 articles were published in 2020 followed by 9 articles in 2017. Besides, only one article was reported in 2010” This is an anecdotal way of describing year of publication, given that 44 studies were included. Please reconsider how to present these data.

Response: This result is rephrased in a more informative way. The detail is displayed in table 2 and description is reserved only for the majority of studies and range of year of publication.

Comment: Tables need legends to explain the acronyms

Response: Legends are added in some tables to avoid confusions. For example “ART” in table 2.

Comment: Figure 3. This is a very unconventional forest plot that does not display pooled estimates and confidence intervals well.

Response: Figure 3 is a sensitivity analysis graph. It, like the funnel plot, serves as a visual guide to publication bias. The graph shows estimates are within the confidence interval limit and removal of studies to minimize publication bias is not required. The sensitivity analysis indirectly supports the risk of bias assessment, which was arbitrarily assessed using NOS.

---

## [Decision Letter · Decision Letter 2]

21 Dec 2022

PONE-D-21-36754R2Prevalence of under-nutrition and associated factors among adults taking antiretroviral therapy in sub-Saharan Africa: A systematic review and meta-analysisPLOS ONE

Dear Dr. Ali,

Thank you for submitting your manuscript to PLOS ONE. After careful consideration, we feel that it has merit but does not fully meet PLOS ONE’s publication criteria as it currently stands. Therefore, we invite you to submit a revised version of the manuscript that addresses the points raised during the review process.

We look forward to receiving your revised manuscript.

Kind regards,

Joel Msafiri Francis, MD, MS, PhD

Academic Editor

PLOS ONE

Reviewers' comments:

Reviewer's Responses to Questions

**Comments to the Author**

1. If the authors have adequately addressed your comments raised in a previous round of review and you feel that this manuscript is now acceptable for publication, you may indicate that here to bypass the “Comments to the Author” section, enter your conflict of interest statement in the “Confidential to Editor” section, and submit your "Accept" recommendation.

Reviewer #4: (No Response)

Reviewer #5: All comments have been addressed

2. Is the manuscript technically sound, and do the data support the conclusions?

Reviewer #4: Partly

Reviewer #5: Partly

3. Has the statistical analysis been performed appropriately and rigorously? 

Reviewer #4: Yes

Reviewer #5: Yes

4. Have the authors made all data underlying the findings in their manuscript fully available?

Reviewer #4: Yes

Reviewer #5: Yes

5. Is the manuscript presented in an intelligible fashion and written in standard English?

Reviewer #4: Yes

Reviewer #5: Yes

6. Review Comments to the Author

Reviewer #4: Thank you for inviting me to review this systematic review on the prevalence of under-nutrition among people living with HIV in sub-Saharan Africa. While this topic is of high interest, and the authors have already improved their manuscript during the revision process, I feel the article is not yet ready for publication, and need further details, especially on the methods, results and discussion. Please see below my point-by-point review:

Title: it seems the review includes adults living with HIV, whatever their ART status, fig5 showing sub-analysis by ART status, I would therefore suggest changing the title to fit better with the review.

Abstract:

- Undernutrition has several definitions (low BMI, low weight, small mid-upper arm circumference, micronutrients deficiencies), this needs to be defined clearly in the abstract, especially as the background talks also about food insecurity which can be confusing. There are many terms used (malnutrition, wasting etc..) in the abstract and throughout the paper, which needs to be harmonized

- Methods section needs to be a bit more detailed (inclusion criteria, study period, study design and data extracted, especially related to the associated factors)

- Results section could start with the number of total records identified with the search strategy before to give the final number.

- Given the heterogeneity of the selected studies, whether in terms of study design, population, context, year etc.., I would be extremely careful while comparing the prevalence by country or years. I would rather focus on the HIV-related associated factors, which can be the more useful to target to further improve the nutritional status of people living with HIV.

Introduction: overall this part lacks a bit of structure and justification

- First paragraph should explain the HIV epidemic among adults, and challenges of HIV care, especially in sub-Saharan Africa, I don’t think the part about food insecurity should be there

- Second paragraph about the relationship between nutrition and HIV: ok

- Third paragraph, before objectives and perspectives, about the knowledge gaps on this topic and why it is important to study them: we need to estimate the burden of under-nutrition among people living with HIV in sub-Saharan Africa and understand what are the main risk factors for this population. If the focus in on adults among ART (which again is not clear in the title and abstract), this has to be better justified too

Methods:

- I don’t understand the comparison/control group here. Most studies selected don’t compare the prevalence of under-nutrition by HIV status, and the following results will not really display this. Does the comparative sample was really a criterion of the search strategy?

- Age threshold at age 16 years also need justification, why include youth for which nutritional and growth outcomes are still evolving greatly compared to adults?

- What was the period of time selected for the searching strategy? It is said until November 2021 but no information about the beginning, only in the title of the figures (2009), what is the justification for this threshold?

- I would suggest creating a table for the search strategy rather than putting it in the text, to better highlight it.

Results: Need further details for the reader to understand and have the full picture

- Table 2: Results should be much more specific and includes for each selected article: gender/sex distribution, ART duration (at least +/- 6 months on ART), the type of setting (urban, semi-urban, rural), median age or age range if available, the associated factors measured (so we know for each pooled analysis how many articles you included). I don’t think the response rate is useful here. Add the prevalence with its 95% confidence interval or interquartile range (fig 3 seems to display it but is not easy to read and I would suggest to delete it).

- Fig5: can this figure be stratified also by duration on ART? (whether +/- 6 months, 1 year or 5 years, depending on your results)

- Fig 6,7 and 8: it is not specific what was the reference group for each study. For example for WHO clinical stage, does all the studies compared stage 3-4 with stage 1-2? What was the comparison groups for CD4 count used to calculate the odds ratio? Same question for age and sex, how were the estimates calculated? If the definition and comparison were heterogeneous between studies I don’t see the point of doing a pooled analysis, this could be a narrative review only for this part, with a table explaining the detailed results.

Discussion: overall this part needs to be reinforced and better structured

- 1st paragraph: should summarised the results of the review first, not giving general messages, better suitable for the introduction of the end of the conclusion

- 2nd paragraph: I don’t understand why comparing the results with a population of pregnant women, this is a specific topic. Also, why discussing the effects of undernutrition on mortality here? This outcome was not taken into account in the review.

- In 3rd paragraph, you are discussing and comparing results by sex but this information is not well highlighted in you own article. What was the pooled prevalence by sex in your review? The comparison with results found on children are not necessary here. This will be more relevant to compare with the estimates of under-nutrition among the general population, to see if people living with HIV are more affected by this, and how much more affected.

- Ref28: why discussing of discrepancy with your study while this review measured food insecurity and not under-nutrition, which are, as you said yourself, two different markers?

- Further in the same paragraph, comparing with studies focused on overweight and obesity seems not relevant as well, unless you incorporate those results in your definition of the study outcomes and describe it.

- You briefly described the difference of prevalence per year in your results, I wonder if it would not be worth it to see and discuss if the prevalence has evolved over time, or at least comparing 2009-2015 and 2015-2021 for example. There might have been very few improvements, which is a result in itself, worthwhile to highlight.

- Other limitations need to be highlighted: heterogeneity and lack of comparative group. You should also discuss the risk of publication bias and how you mitigated it.

Reviewer #5: I have reviewed manuscript titled "Prevalence of under-nutrition and associated factors among adults taking antiretroviral therapy in sub-Saharan Africa: A systematic review and meta-analysis"

General comments : Few areas in the manuscript require proof reading. Correct the sentence on page 4, paragraph 1, " For example, a study......times higher odds.............."

If possible remove table 1 and you can just define the PICO questions by text. For this case, P can stand for population, I intervention and C: Comparison..... Also correct the sentence under population to read 16 years instead of 16 year... Also be consistent, is it grey literature or gray literature?

Under Effect measures : Make sure you write that sentence in past tense "We included......"

Methodology : Properly explain why I2 is high

Discussion: Page 13, correct those 2 sentences " Conversely.........." " The study used....

The second paragraph towards the end, you are discussing about children. This article is on adults.

Thank you

7. PLOS authors have the option to publish the peer review history of their article (what does this mean?). If published, this will include your full peer review and any attached files.

Reviewer #4: No

Reviewer #5: No

---

## [Author Response · Author response to Decision Letter 2]

30 Dec 2022

Response to reviewers #4

First, I would like to thank the reviewers for taking the time to provide us with invaluable comments that enrich the quality of the manuscript.

Comment: Title: it seems the review includes adults living with HIV, whatever their ART status, fig5 showing sub-analysis by ART status, I would therefore suggest changing the title to fit better with the review.

Response: Yes, the manuscript includes adults living with HIV irrespective of their ART status, and the pooled prevalence is reported in the document. However, subgroup analysis by different relevant characteristics is common in order to better understand the results by different categories. The sub-group analysis was made to look into whether taking ART has made a difference in nutritional status. There is evidence that taking ART improves nutritional status. Sub-group analysis is made for description, not inference. We believe sub-group analysis by different variables will not demand changing the title.

Abstract:

Comment- Undernutrition has several definitions (low BMI, low weight, small mid-upper arm circumference, micronutrients deficiencies), this needs to be defined clearly in the abstract, especially as the background talks also about food insecurity which can be confusing. There are many terms used (malnutrition, wasting etc..) in the abstract and throughout the paper, which needs to be harmonized

Response: Though malnutrition also includes overnutrition, in this study and commonly, malnutrition, undernutrition, underweight, wasting, and malnourishment are all used in the same context. All of the studies included in this meta-analysis defined malnutrition as low BMI (less than or equal to 18.5 kg/m2), as indicated in the main text. Food security was mentioned as one of the causes of malnutrition in SSA. However, as it is not the scope of this study, we removed it from the statement. We put the operational definition in the first statement in brackets.

Comment: Methods section needs to be a bit more detailed (inclusion criteria, study period, study design and data extracted, especially related to the associated factors)

Response: Accepted and we add some details in methods part.

Comment: Results section could start with the number of total records identified with the search strategy before to give the final number.

Response: Different articles present the results in different ways in their abstracts. Overall, we used the PRISMA 2020 Abstracts Checklist to structure our abstract. Because the details are described in text and using a flow chart in the main result section, we prefer not to present the number of total records here.

Comment: Given the heterogeneity of the selected studies, whether in terms of study design, population, context, year etc.., I would be extremely careful while comparing the prevalence by country or years. I would rather focus on the HIV-related associated factors, which can be the more useful to target to further improve the nutritional status of people living with HIV.

Response: Thank you for the concern. Sub-group analysis is used in meta-analysis to describe phenomena by different segments rather than for inference. It is also used to identify the source of heterogeneity. Inference is made based on the major objective of the study, i.e., the pooled estimate of prevalence and associated factors. Therefore, we removed the sub-group result from the abstract as it was not our major finding. Additionally, we avoid comparison statements like "the highest" for the sub-group results presented in the main text.

Introduction: overall this part lacks a bit of structure and justification

Comment: First paragraph should explain the HIV epidemic among adults, and challenges of HIV care, especially in sub-Saharan Africa, I don’t think the part about food insecurity should be there.

Response: We really appreciate the comment on the structural organization of the contents presented in the introduction. A number of literatures on food insecurity were added as per previous reviewers’ comments on why the "sub-Saharan Africa region" was selected as the study area. It was to show the high burden of food insecurity and HIV transmission in SSA, which puts these patients at high risk for malnutrition. However, we abridged the statements on food insecurity and brought them to the third paragraph, where the evidence gaps are presented. Similarly, we added some statements describing the epidemiology of HIV among adults and the challenges of HIV care in SSA as background statements in the first paragraph.

Comment: Third paragraph, before objectives and perspectives, about the knowledge gaps on this topic and why it is important to study them: we need to estimate the burden of under-nutrition among people living with HIV in sub-Saharan Africa and understand what are the main risk factors for this population. If the focus in on adults among ART (which again is not clear in the title and abstract), this has to be better justified too.

Response: In the third paragraph, we presented a summary of previous key findings related to the topic and showed the evidence gap and how our study contributed to filling it. The reason for categorizing based on ART status is that there is evidence that indicates taking ART improves nutritional status, and sub-group analysis by ART status will enable us to appreciate the proportion separately.

Methods:

Comment- I don’t understand the comparison/control group here. Most studies selected don’t compare the prevalence of under-nutrition by HIV status, and the following results will not really display this. Does the comparative sample was really a criterion of the search strategy?

Response: In cross-sectional studies, the comparator group is identified by context or assumption. The results are interpreted under this assumption. However, in case-control and cohort studies, the control groups are those who are HIV-negative adults. The comparative sample was not a criterion of the search strategy but a context for interpretation.

Comment: Age threshold at age 16 years also need justification, why include youth for which nutritional and growth outcomes are still evolving greatly compared to adults?

Response: The lowest age categories for adults in the included studies were 15 years, 16 years, 18 years, and 20 years. Adults are defined as people over the age of "15 years" by the UNAIDS definition. It is corrected in the PICO. https://www.unaids.org/en/resources/fact-sheet

Comment: What was the period of time selected for the searching strategy? It is said until November 2021 but no information about the beginning, only in the title of the figures (2009), what is the justification for this threshold?

Response: Accepted. We include the commencement period. The reason was that there was a previous meta-analysis study on the prevalence of undernutrition among women living with HIV published in 2008. To avoid redundancy and add evidence to existing knowledge, we took published studies starting in 2009 and included both adult males and females. https://bmcpublichealth.biomedcentral.com/articles/10.1186/1471-2458-8-226

Comment- I would suggest creating a table for the search strategy rather than putting it in the text, to better highlight it.

Response: It is possible to present search strategy in table. 

Results: Need further details for the reader to understand and have the full picture

Comment- Table 2: Results should be much more specific and includes for each selected article: gender/sex distribution, ART duration (at least +/- 6 months on ART), the type of setting (urban, semi-urban, rural), median age or age range if available, the associated factors measured (so we know for each pooled analysis how many articles you included). I don’t think the response rate is useful here. Add the prevalence with its 95% confidence interval or interquartile range (fig 3 seems to display it but is not easy to read and I would suggest to delete it).

Response: We include some of the suggested variables, like 95% CI and mean/median age, but inserting all variables will overwhelm the table and make it unattractive. Studies on ART duration were conducted in both categories, i.e., no study was conducted on either the -6 or +6 month separately. Most importantly, the data on the associated factors, including ART duration, are extracted on a separate Excel sheet. I will attach it as a supplementary file for further clarity. The response rate is added to show the credibility of the reported estimate. The forest plot for each associated factor would tell us the number of studies included there. For better understanding, we created a table that summarizes the points related to the associated factors (Table 4).

Figure 3 is a sensitivity analysis graph. It, like the funnel plot, serves as a visual guide to publication bias. The graph shows the effect estimates (indicated by small circles) are within the confidence interval limit, and no outlier is detected. Thus, the removal of studies to minimize publication bias is not required. The sensitivity analysis indirectly supports the risk of bias assessment, which was arbitrarily assessed using NOS. We believe adding this graph will enhance the quality of this manuscript.

Comment: Fig5: can this figure be stratified also by duration on ART? (whether +/- 6 months, 1 year or 5 years, depending on your results)

Response: No, it is not possible. In order to stratify by the duration of ART, individual studies should study patients with a specific category of treatment duration. Studies were not conducted with a certain limit on treatment duration. Most importantly, the pooled estimate of the adjusted odds ratio between "duration of ART" and "undernutrition" is presented in the "results" section and was 2.68.

Comment: Fig 6,7 and 8: it is not specific what was the reference group for each study. For example, for WHO clinical stage, does all the studies compared stage 3-4 with stage 1-2? What was the comparison groups for CD4 count used to calculate the odds ratio? Same question for age and sex, how were the estimates calculated? If the definition and comparison were heterogeneous between studies, I don’t see the point of doing a pooled analysis, this could be a narrative review only for this part, with a table explaining the detailed results.

Response: We appreciate this outlook. We did not combine heterogenous comparison groups. We include the reference group that the majority of the studies used. We exclude articles that report on a completely different reference group. This is the major problem that researchers face during meta-analysis of determinants (associated factors) with outcome variables. We used the reported adjusted odds ratio to determine the strength of the association. We prefer to present a short summary of the associated factors and comparison group using a table format. (Table 4).

Discussion: overall this part needs to be reinforced and better structured

Comment: 1st paragraph: should summarized the results of the review first, not giving general messages, better suitable for the introduction of the end of the conclusion.

Response: We generally agree with the comment. We also understand that the first paragraph in the discussion summarizes major concepts outlined in the introduction and provides a logical linkage of the results with the original question or objectives. We make amendments based on the above.

Comment: 2nd paragraph: I don’t understand why comparing the results with a population of pregnant women, this is a specific topic. Also, why discussing the effects of undernutrition on mortality here? This outcome was not taken into account in the review.

Response: The comment is fully accepted. This was added due to the scarcity of similar meta-analysis studies for comparison with our findings. However, rather than compare with a different population and outcome variable, we decided to remove the comparators and mention the shortage of evidence under the limitation of study section.

Comment: In 3rd paragraph, you are discussing and comparing results by sex but this information is not well highlighted in you own article. What was the pooled prevalence by sex in your review? The comparison with results found on children are not necessary here. This will be more relevant to compare with the estimates of under-nutrition among the general population, to see if people living with HIV are more affected by this, and how much more affected.

Response: We presented the sex, age, ART duration, WHO clinical stage, and CD4 count in the result section under the sub-heading “Factors associated with under-nutrition”. Describing the prevalence by subgroup is different from determining the strengths of association between the factors and outcome variable. The pooled adjusted odds ratio is separately estimated for each above-mentioned factor. For clarity we will attach the data extraction excel sheet for all factors as a supplementary file. 

Comment: Ref28: why discussing of discrepancy with your study while this review measured food insecurity and not under-nutrition, which are, as you said yourself, two different markers?

Response: Again, we also removed this section. It was inserted as food insecurity is closely related and leads to under-nutrition in majority of the studies. Similarly, this was used due to shortage of meta-analysis studies exactly carried out on under-nutrition. 

Comment: Further in the same paragraph, comparing with studies focused on overweight and obesity seems not relevant as well, unless you incorporate those results in your definition of the study outcomes and describe it.

Response: Again, this comment is accepted. It was used for the above similar reason. 

Comment: You briefly described the difference of prevalence per year in your results, I wonder if it would not be worth it to see and discuss if the prevalence has evolved over time, or at least comparing 2009-2015 and 2015-2021 for example. There might have been very few improvements, which is a result in itself, worthwhile to highlight. 

Response: We appreciate you suggesting this perspective for interpretation, and we will include it in the result and discussion.

Comment: Other limitations need to be highlighted: heterogeneity and lack of comparative group. You should also discuss the risk of publication bias and how you mitigated it.

Response: High heterogeneity is expected in the meta-analysis of proportions reported from different countries. The source of heterogeneity was explored by sub-group analysis but not identified. The inter-country difference in population characteristics is obvious. It is also mentioned in the document under the sub-heading of quality appraisal results as "this result should be interpreted cautiously." We will include it in the limitation section. Regarding the publication bias, studies look symmetrically distributed in the funnel plot (Figure 2) but not by the Eggers statistical test. The sensitivity analysis graph (Figure 3) did not show outlier studies to be removed from the analysis. Trim and fill analysis is also another method, but it is not recommended as it provides artificial estimates. Additionally, the small study size effect is not the problem of this study, as 44 articles have been included in the analysis. Therefore, it is acceptable to run the analysis based on available studies, and publication bias will not be the limitation of this study.

Response to reviewer #5

General comments: Few areas in the manuscript require proof reading. Correct the sentence on page 4, paragraph 1, " For example, a study......times higher odds.............."

Response: Okay, accepted. 

Comment: If possible, remove table 1 and you can just define the PICO questions by text. For this case, P can stand for population, I intervention and C: Comparison..... Also correct the sentence under population to read 16 years instead of 16 year... Also be consistent, is it grey literature or gray literature?

Response: Accepted. PICO questions can be formatted by texts using bullets. Though the words “grey” and “gray” can be used interchangeably we will use one of the terms throughout the paper.

Comment: Under Effect measures: Make sure you write that sentence in past tense "We included......"

Response: Accepted.

Comment: Methodology: Properly explain why I2 is high.

Response: Still, the source is not identified, but high heterogeneity is expected when combining proportions reported from different countries. The only thing we can do is add the statement "The result should be interpreted cautiously" and mention high heterogeneity under the limitation of the study section.

Comment: Discussion: Page 13, correct those 2 sentences " Conversely.........." " The study used....

Response: Okay, Accepted.

Comment: The second paragraph towards the end, you are discussing about children. This article is on adults.

Response: The paragraph on children was added due to scarcity of available similar studies on adults. However, we delete it as it is beyond the scope of our study defined in the PICO.

---

## [Decision Letter · Decision Letter 3]

12 Mar 2023

Prevalence of under-nutrition and associated factors among adults taking antiretroviral therapy in sub-Saharan Africa: A systematic review and meta-analysis

PONE-D-21-36754R3

Dear Dr. Ali,

We’re pleased to inform you that your manuscript has been judged scientifically suitable for publication and will be formally accepted for publication once it meets all outstanding technical requirements.

Kind regards,

Joel Msafiri Francis, MD, MS, PhD

Academic Editor

PLOS ONE

Additional Editor Comments (optional):

Please format the figures - and would be helpful to drop Figure 3. It is not informative. It would would be helpful to proof read the English grammar prior to publication. 

Reviewers' comments:

Reviewer's Responses to Questions

**Comments to the Author**

1. If the authors have adequately addressed your comments raised in a previous round of review and you feel that this manuscript is now acceptable for publication, you may indicate that here to bypass the “Comments to the Author” section, enter your conflict of interest statement in the “Confidential to Editor” section, and submit your "Accept" recommendation.

Reviewer #5: All comments have been addressed

2. Is the manuscript technically sound, and do the data support the conclusions?

Reviewer #5: Yes

3. Has the statistical analysis been performed appropriately and rigorously? 

Reviewer #5: Yes

4. Have the authors made all data underlying the findings in their manuscript fully available?

Reviewer #5: Yes

5. Is the manuscript presented in an intelligible fashion and written in standard English?

Reviewer #5: Yes

6. Review Comments to the Author

Reviewer #5: Thank you for responding to reviewers comments. Please verify that UNAIDS identify adults as those aged above 15.

7. PLOS authors have the option to publish the peer review history of their article (what does this mean?). If published, this will include your full peer review and any attached files.

Reviewer #5: No

---

## [Editor Report · Acceptance letter]

16 Mar 2023

PONE-D-21-36754R3 

Prevalence of undernutrition and associated factors among adults taking antiretroviral therapy in sub-Saharan Africa: A systematic review and meta-analysis 

Dear Dr. Seid:

I'm pleased to inform you that your manuscript has been deemed suitable for publication in PLOS ONE. Congratulations! Your manuscript is now with our production department. 

Kind regards, 

on behalf of

Dr. Joel Msafiri Francis 

Academic Editor

PLOS ONE